# Hormetic Responses of Photosystem II in Tomato to *Botrytis cinerea*

**DOI:** 10.3390/plants10030521

**Published:** 2021-03-10

**Authors:** Maria-Lavrentia Stamelou, Ilektra Sperdouli, Ioanna Pyrri, Ioannis-Dimosthenis S. Adamakis, Michael Moustakas

**Affiliations:** 1Section of Botany, Department of Biology, National and Kapodistrian University of Athens, GR-15784 Athens, Greece; marilorast@yahoo.gr (M.-L.S.); iadamaki@biol.uoa.gr (I.-D.S.A.); 2Institute of Plant Breeding and Genetic Resources, Hellenic Agricultural Organization–Demeter, Thermi, GR-57001 Thessaloniki, Greece; ilektras@bio.auth.gr; 3Section of Ecology & Systematics, Department of Biology, National and Kapodistrian University of Athens, GR-15784 Athens, Greece; ipyrri@biol.uoa.gr; 4Department of Botany, Aristotle University of Thessaloniki, GR-54124 Thessaloniki, Greece

**Keywords:** biotic stress, reactive oxygen species (ROS), photoprotection, non-photochemical quenching (NPQ), hormesis, hydrogen peroxide (H_2_O_2_), photosynthesis, toxicity, stress defense, chlorophyll fluorescence imaging

## Abstract

*Botrytis cinerea*, a fungal pathogen that causes gray mold, is damaging more than 200 plant species, and especially tomato. Photosystem II (PSII) responses in tomato (*Solanum lycopersicum* L.) leaves to *Botrytis cinerea* spore suspension application were evaluated by chlorophyll fluorescence imaging analysis. Hydrogen peroxide (H_2_O_2_) that was detected 30 min after *Botrytis* application with an increasing trend up to 240 min, is possibly convening tolerance against *B. cinerea* at short-time exposure, but when increasing at relative longer exposure, is becoming a damaging molecule. In accordance, an enhanced photosystem II (PSII) functionality was observed 30 min after application of *B. cinerea*, with a higher fraction of absorbed light energy to be directed to photochemistry (Φ*_PSΙΙ_*). The concomitant increase in the photoprotective mechanism of non-photochemical quenching of photosynthesis (NPQ) resulted in a significant decrease in the dissipated non-regulated energy (Φ*_NO_*), indicating a possible decreased singlet oxygen (^1^O_2_) formation, thus specifying a modified reactive oxygen species (ROS) homeostasis. Therefore, 30 min after application of *Botrytis* spore suspension, before any visual symptoms appeared, defense response mechanisms were triggered, with PSII photochemistry to be adjusted by NPQ in a such way that PSII functionality to be enhanced, but being fully inhibited at the application spot and the adjacent area, after longer exposure (240 min). Hence, the response of tomato PSII to *B. cinerea*, indicates a hormetic temporal response in terms of “stress defense response” and “toxicity”, expanding the features of hormesis to biotic factors also. The enhanced PSII functionality 30 min after *Botrytis* application can possible be related with the need of an increased sugar production that is associated with a stronger plant defense potential through the induction of defense genes.

## 1. Introduction

*Botrytis cinerea*, a fungal pathogen that causes gray mold, is damaging more than 200 plant species, especially tomato, cucumber, strawberry, potato, and ornamental plants [1,2]. *B. cinerea* releases ethylene to accelerate leaf wither, decreasing photosynthetic efficiency and affecting plant production resulting in huge economic losses [2,3]. This pathogen kills plant tissues previous to feeding on them, and uses a variety of toxic molecules to decompose the host cells [4,5]. Symptoms initial develop on infected tissues as limited lesions that subsequently become necrotic and spread to other tissues [5]. Thus, lesion development induced by the necrotrophic fungus *B. cinerea* influences growth performance of tomato plants [6]. Sugars produced by plants, although beneficial for the pathogen as nutrient source, are often associated with stronger plant defense potential, through their involvement in the induction of phenylpropanoids and/or defense genes [7]. Infection by *B. cinerea* induces increased respiration on the infected tissues, leading to a severe drop in the O_2_ concentration in an otherwise fully aerobic leaf [8].

In plant cellular metabolism, and especially in chloroplasts, reactive oxygen species (ROS) are continuously produced at basal levels that are incapable to cause damage, as they are being scavenged by different antioxidant mechanisms [9,10,11,12]. However, under most biotic or abiotic stresses an increased production of ROS occurs that can lead to oxidative stress if it is not scavenged by enzymatic or non-enzymatic antioxidants [11,12,13,14,15]. However, ROS derived from the chloroplasts also play a role in plant resistance against *B. cinerea* [15].

ROS, such as superoxide anion radical (O_2_**^•^****^−^**), hydrogen peroxide (H_2_O_2_), and singlet oxygen (^1^O_2_), produced in chloroplasts play dual roles as they generate oxidative stress and also confer essential biological function as redox signaling in response to biotic and abiotic stress conditions [14,15,16]. The role of chloroplast antioxidants, that often have overlying or interrelating functions, is not to totally eliminate ROS, but rather to achieve a suitable balance between production and removal so that to counterpart photosynthetic function, permitting an effective diffusion of signals to the nucleus and adjusting a plethora of physiological functions [13,16,17]. Leaf image analysis is also used as an objective and accurate alternative method to quantify ROS production at the affected leaf areas, and has proven to be a useful tool in plant science [15].

Photosynthesis is a fundamental process in plant physiology, and its regulation plays a central role in plant defense against biotic and abiotic stresses [18,19]. Photosynthesis studies seem incomplete without some chlorophyll fluorescence data, since minor modifications in plant metabolism make this technique appropriate to provide insight into plant-stress interaction [18,19,20,21,22,23,24]. Chlorophyll fluorescence analysis has been extensively used to uncover perturbations on the photosynthetic efficiency due to nutrient deficiency, drought stress, soil salinity, metal toxicity, wastewater toxicity, pathogenic diseases, etc. [25,26,27,28,29,30,31,32,33]. The development of the method of chlorophyll fluorescence imaging analysis (CF-IA) permits the revealing of the leaf spatiotemporal photosynthetic heterogeneity at the total leaf surface, that is ignored when measuring chlorophyll fluorescence with the classical point measurement techniques [18,20,21,22,23]. CF-IA is a highly effective non-invasive method for the estimation of the inhibition or damage in the PSII electron transfer process, serving as an extremely sensitive indicator of photosynthetic efficiency under both biotic and abiotic stress conditions [24,34,35,36,37,38,39]. The fact that biotic stresses are typically heterogeneous, being spatiotemporal, renders chlorophyll fluorescence imaging apparatuses powerful tools to study the impact of biotic stress on leaf photosynthetic performance [37].

Plants respond to biotic and abiotic stresses by a plethora of mechanisms [13,40]. A low-level of stress or a short exposure duration has been frequently referred to stimulate plant performance [13,20,40,41,42,43,44]. This enhancement can be achieved through a basal level of ROS [9,10,13,16,45], that is being regulated by the photoprotective mechanism of photosynthesis, the non-photochemical quenching (NPQ) [23,43,46]. NPQ under stress conditions decreases electron transport rate (ETR) to avoid ROS formation [35,47]. ROS production that can occur through PSII damage can prevent the repair of PSII reaction centres [35,48]. Dose–response studies are proposing hormesis as a widespread phenomenon usual in nature and independent of the kind of stressor, the physiological process or the organism it occurs [13,40,43,44,49].

In the present work we used CF-IA to quantify the impact of *B. cinerea* spore suspension application on photosystem II (PSII) in tomato leaves, combined with ROS detection at the inoculated leaf area. We monitored the time-lapsed PSII functionality and try to gain an insight into the mechanisms that play a role in plant defense response to the fungal pathogen.

## 2. Results

### 2.1. Visible Symptoms of Botrytis cinerea Spore Application on Tomato

Visual damage on tomato leaflets could be detected only after 24 h of the 20 μL spore suspension application (10^5^–10^7^ spores/mL) (Figure 1a), with the apparent hyphae formation to be also observed (Figure 1b). Leaflet damage appeared as a discoloration of the leaf epidermis (white spot) (arrow in Figure 1a).

### 2.2. Hydrogen Peroxide Detection after Botrytis cinerea Spore Application

At 30 min after *B. cinerea* spore suspension application, H_2_O_2_ production was detected at the drop’s application area only (arrowhead in Figure 1c), but with time, H_2_O_2_ production gradually increased and spread out (Figure 1d). Thus, after 240 min, DCF-DA signal intensified in the application area (arrowhead in Figure 1d) and H_2_O_2_ detection expanded to the leaf veins as well (arrow in Figure 1d).

### 2.3. Allocation of Absorbed Light Energy at PSII before and after Spore Application

We estimated the fraction of the absorbed light energy that is used for photochemistry (Φ_PSΙΙ_), is lost by regulated heat dissipation (Φ_NPQ_), and that of non-regulated energy loss (Φ*_NO_*), that add up to unity [35]. The effective quantum yield of PSII photochemistry (Φ_PSII_) 30 min after spore suspension application increased significantly compared to control, while 120 min after spore suspension application it did not differ compared to control, at both LL and HL treatments. Φ_PSII_ decreased significantly compared to control, 240 min after spore suspension application, at both LL and HL (Figure 2). The quantum yield of regulated non-photochemical energy loss in PSII (Φ_NPQ_) increased significantly up to 120 min after spore suspension application, but later on decreased to control level at LL (Figure 3a), while at HL treatment there was no significant difference compared to control at all measurements after spore suspension application (Figure 3b).

Due to the increase of Φ_NPQ_ at 30 and 120 min after spore suspension application, the quantum yield of non-regulated energy loss in PSII (Φ_NO_) decreased compared to control, but increased significantly 240 min after spore suspension application at LL (Figure 4a). At HL, the pattern was similar to LL, with the exception that of 120 min after spore suspension application, where ΦNO was at the same level with control plants (Figure 4b).

### 2.4. Photoprotective Dissipation of Excitation Energy as Heat (NPQ)

The non-photochemical chlorophyll fluorescence quenching (NPQ) increased significantly up to 120 min after spore application, but later on decreased to control level at LL (Figure 5a). At HL treatment, there was no significant difference compared to control, at 30 and 120 min after spore application, while at 240 min after spore application, NPQ was significantly lower than control (Figure 5b).

### 2.5. Chlorophyll a Fluorescence Images

At 30 min after application of *B. cinerea* suspension, Φ_PSII_ decreased in the direct vicinity of the spore application area, compared to control values (arrow in Figure 6). At the same time at the surrounding area and the rest of the leaflet, an increased Φ_PSII_ was observed, having as a result a higher Φ_PSII_ value at the whole leaflet compared to control. At 240 min after spore application, the effective quantum yield of PSII photochemistry was severely affected at the whole leaflet area, being totally interrupted (Φ_PSII_ = 0) at the application spot and the adjacent area, and on most of the leaflet, with a high photosynthetic heterogeneity to be observed (Figure 6).

Following the pattern of Φ_PSII_, the redox state of the plastoquinone pool, that is a measure of the number of open PSII reaction centers (*q*_p_), 30 min after spore suspension application increased at the whole leaflet (54% open), compared to control (44% open) (Figure 7). In accordance to Φ_PSII_ pattern, at 240 min after spore application, most reaction centers at the application spot and the adjacent area, were completely closed (*q*_p_ = 0), and also on most of the leaflet, with only a 5% of reaction centers to remain open at the whole leaflet (Figure 7).

## 3. Discussion

Plant pathogens are divided into biotrophics, that attack the host cells conserving host viability and gaining nutrients from living cells, and necrotrophics, that get their nutrients by killing the host plant [8]. *Botrytis cinerea* is a necrotrophic plant pathogen causing gray mold disease on several crop plant species, producing enormous damage in crop production [4]. Thus, there is increasing interest in the mechanism(s) utilized by plants to counteract infection by this fungus [8].

The necrotrophic pathogen *B. cinerea* is able to induce ROS generation, especially H_2_O_2_, in a plethora of plant species [50,51], either directly or indirectly (e.g., via the toxin botrydial) [52]. In our experimental set up, H_2_O_2_ was detected 30 min after spore suspension application and gradually increased (Figure 1). The question that arises is on the role of this H_2_O_2_ production. One possible explanation proposed from various researchers was that the H_2_O_2_ produced could directly fight off the fungi [51]. However, this hypothesis was abandoned since H_2_O_2_ production does not affect fungi development, given that *B. cinerea* poses effective ROS-detoxification systems, while at the same time *B. cinerea* can even contribute to ROS increase by producing its own ROS [50,51]. The rapid accumulation of H_2_O_2_ observed (Figure 1), is a widespread defense mechanism of higher plants against pathogen attack [50].

Non-photochemical chlorophyll fluorescence quenching (NPQ), the key photoprotective process in plants that dissipates excess light energy as heat and protects photosynthesis [11,12,53,54,55,56], increased significantly up to 120 min after spore suspension application, but later on decreased to control level (Figure 5a). Similarly, NPQ increased in ice leaves in the surrounding area of the infection by *B. cinerea* [24]. After 240 min of *B. cinerea* spore suspension application, NPQ at HL was significantly lower than the control (Figure 5b). This indicates that when *Botrytis* application was combined with HL, the photoprotective mechanism was no longer buffering the excess light stress levels, indicating an imbalance between energy supply and demand [57,58,59,60]. This, results in increased ^1^O_2_ and H_2_O_2_ production [61] as we observed (Figure 1b and Figure 4b) respectively. Non-photochemical quenching is a major component of the systemic acquired acclimation and systemic acquired resistance which is tightly related to ROS [10,56]. Chloroplasts through the operation of photosynthesis play an important role as redox sensors of environmental conditions and elicit acclimatory or stress defense responses [62,63]. Tomato leaflets after *B. cinerea* spore suspensions application show an increased capacity to keep quinone (QA) oxidized, thus, to have a higher fraction of open PSII reaction centers (*q*_p_) compared to controls (Figure 7), indicating an enhanced PSII functionality.

At 30 min after spore suspensions application, a decreased Φ_PSII_ in the direct vicinity of the application spot compared to control values was detected (Figure 6), while at the surrounding area and the rest of the leaflet, an increased Φ_PSII_ was observed, specifying that the biotic stress was signaled to the rest of the leaflet regions, distant from the spore application area, having as a result a higher Φ_PSII_ value at the whole leaflet compared to control. It has been frequently shown that H_2_O_2_ diffuses through leaf veins to act as a long-distance molecule, triggering the stress defense response in plants [9,12,14,16,23]. An inhibition of photosynthesis by decreased Φ_PSII_ values in the direct vicinity of the *B. cinerea* infection sites of tomato leaflets was recorded [64], but, with no alterations in the primary metabolism in the rest of the leaf tissue [64]. Decays in the effective quantum yield of PSII photochemistry (Φ_PSII_) and increases in NPQ values before any visual symptoms appeared were observed in cashew seedlings inoculated with *Lasiodiplodia theobromae* [65].

The enhancement of Φ_PSII_ 30 min after *B. cinerea* spore application (Figure 2) and the concurrent increase of the regulated non-photochemical energy loss in PSII (Φ_NPQ_) (Figure 3), resulted in a significant decrease in the quantum yield of non-regulated energy loss in PSII (Φ_NO_) (Figure 4). Φ_NO_ comprises of chlorophyll fluorescence internal conversions and intersystem crossing, that results to ^1^O_2_ formation via the triplet state of chlorophyll (^3^chl*) [53,66,67,68]. Thus, after application of *B. cinerea* on tomato leaflets, the decreased Φ_NO_ indicates decreased ^1^O_2_, that is considered as a highly damaging ROS produced by PSII [14,69,70,71,72]. However, 240 min after *B. cinerea* application, the increased Φ_NO_ suggests high levels of ^1^O_2_ (Figure 4) that could act synergistically with the high H_2_O_2_ level (Figure 1d), in chloroplast damage [15,73]. *Botrytis cinerea* is a necrotrophic fungus that produces constantly toxic compounds which ultimately cause cell death [4]. Then, the fungus feeds on the dead tissue, thus, causing noticeable necrotic lesions [4].

While ^1^O_2_ creation via ^3^chl* decreased 30 min after spore application compared to control (Figure 4a), H_2_O_2_ was detected to appear 30 min after spore application (Figure 1c), and gradually to increase (Figure 1d). Hydrogen peroxide levels were positively correlated with disease severity [74], and low hydrogen peroxide levels appear to be the best indicator for leaf resistance to *B. cinerea* in strawberry leaves [74], while, chloroplast-generated ROS play a major role in *B. cinerea* –induced leaf damage [73], that eventually results in cell death (necrosis) [4]. As ROS formation by energy transfer (^1^O_2_), and electron transport (H_2_O_2_) is simultaneous, it seems likely that their signaling pathways occasionally antagonize each other [13]. A low increase in ROS level is considered as favorable for activating defense responses [13,50,75,76], but an excessive ROS level is damaging to PSII functionality [9,13], which seems to be the case in our experimentation.

*Botrytis cinerea* that has been placed second in rank order of the top 10 fungal plant pathogens [77] can develop microscopic infections in epidermal cells of leaves that remain hidden for a certain period but after sporulation they convert in sources of primary inoculum and spread to other tissues [74].

The enhanced PSII functionality 30 min after *Botrytis* application can possible be related with the need of an increased sugar production that is associated with a stronger plant defense potential through the induction of defense genes [7]. However, increased levels of soluble sugars have been shown to support *B. cinerea* growth in tomato leaves [6]. In *Rosa chinensis* leaves, exogenous jasmonate (JA) spraying was shown to be essential for inducing defense response against *B. cinerea* [5]. Plant defenses can be activated by ROS [50], or by elicitor molecules such as chitosan [78], and γ-aminobutyric acid (GABA) [79], that are able to induce resistance mechanisms against *B. cinerea*.

Photosystem II responses of tomato leaflets to *B. cinerea* inoculation can be described as a time-dependent hormetic response (Figure 8) in terms of “stress defense response” and “toxicity” [76,80]. Activation by a low-dose effect or short-time exposure is a common phenomenon that is associated with the term “hormesis”, typically indicating a positive biological response [13,44,55,81,82,83,84,85,86]. There are published studies reporting a time-dependent hormesis [55,82], but it is only recently that analysis has been made regarding the features of hormesis as a function of time [13,81,82], and according to our data, under biotic stress.

## 4. Materials and Methods

### 4.1. Plant Material and Growth Conditions

Tomato (*Solanum lycopersicum* L.) plants (20 cm in height) were obtained from the market and transported to a growth chamber with 20 ± 1/18 ± 1 °C day/night temperature, 10-h photoperiod with photosynthetic photon flux density (PPFD) 220 ± 20 μmol quanta m^−2^ s^−1^ and relative humidity 50 ± 5/60 ± 5% day/night.

### 4.2. Pathogen Culture and Spore Suspension Application

*Botrytis cinerea* ATHUM 4850 obtained from the ATHUM Culture Collection of Fungi of the National and Kapodistrian University of Athens Mycetotheca was used for tomato leaf spore application and was cultured on a solid nutrient medium (Potato Dextrose Agar, PDA, BD Difco, Oxford, UK) containing 0.5 mL L^−1^ lactic acid 1N at 23 °C until sporulation. Sporulated cultures were transferred in 100 mL sterilized distilled water and gently vortexed for spore release.

Tomato 5th leaflets were detached and laid on Petri dishes containing two sterilized blotting filter papers (Whatman, Schleicher and Schuell, Ottawa, Canada), wetted with sterile water. A drop (20 μL) of *B. cinerea* aqueous spore suspension (10^5^–10^7^ spores/mL) was applied on tomato leaflets with a micropipette [87] as shown on Figure 1a, while leaves treated with a drop (20 μL) of sterile water (without *B. cinera* spores) were considered as controls.

### 4.3. Hydrogen Peroxide Imaging Detection

H_2_O_2_ imaging detection, in control, and at 30, 120, and 240 min after spore suspension application, was performed as described earlier [12]. Briefly, inoculated and non-inoculated tomato leaflets were incubated with 25 μM 2′, 7′-dichlorofluorescein diacetate (DCF-DA, Sigma-Aldrich, Chemie GmbH, Schnelldorf, Germany) for 30 min in the dark, to visualize H_2_O_2_ production.many

Inoculated and control tomato leaflets were observed under a Zeiss Axioplan epifluoresence microscope equipped with an AxioCam MRc 5 digital camera. Photographs were obtained with the ZEN 2 version software according to the manufacturer’s instructions.

### 4.4. Chlorophyll Fluorescence Imaging Analysis

An *Imaging PAM M-Series* system (Heinz Walz Instruments, Effeltrich, Germany) was used for the chlorophyll fluorescence measurements of the dark adapted (15 min) control and infected tomato leaflets as described in detail previously [39]. Measurements were conducted in leaves treated with a drop (20 μL) of sterile water (without *B. cinera* spores, control, 0 min) and after 30, 120 and 240 min of *B. cinerea* aquatic spore suspension application. The light intensities that were used for the photosynthetic efficiency measurements were 230 μmol photons m^−2^ s^−1^, a low light (LL) intensity similar to the growth light, a moderate light (ML) intensity at 640 μmol photons m^−2^ s^−1^, and a high light (HL) one, at 900 μmol photons m^−2^ s^−1^. Representative areas of interest (AOIs) were selected in each leaflet so as to cover the whole leaflet area. The chlorophyll fluorescence parameters that were measured together with their definitions are shown in Table 1. Color-coded images of the effective quantum yield of PSII photochemistry (Φ_PSΙΙ_) and the redox state of the plastoquinone pool (*q*_p_), obtained at ML intensity at 640 μmol photons m^−2^ s^−1^, are also presented.

### 4.5. Statistics

Statistically significant differences among the means were determined, from three independent treatments with each treatment consisting of three leaflets from three different tomato plants, using two-way ANOVA. Means (± SD) were considered statistically different at a level of *p* < 0.05.

## 5. Conclusions

In the present study, 30 min after application of *Botrytis* spore suspension in tomato leaflets, before any visual symptoms appeared, defense response mechanisms were triggered, with light energy use to be adjusted by NPQ in a such way that PSII functionality to be enhanced. The underlying mechanism involved was possible activated by H_2_O_2_, that was detected 30 min after *Botrytis* application with an increasing trend up to 240 min. This, is possibly convening tolerance against *B. cinerea* at short-time exposure, but at relatively longer exposure time when H_2_O_2_ is increasing, it is becoming a damaging molecule. Hence, the response of tomato PSII to *B. cinerea*, indicates a hormetic temporal response, in terms of “stress defense response” and “toxicity”, expanding the features of hormesis to biotic factors also.

## Figures and Tables

**Figure 1 plants-10-00521-f001:**
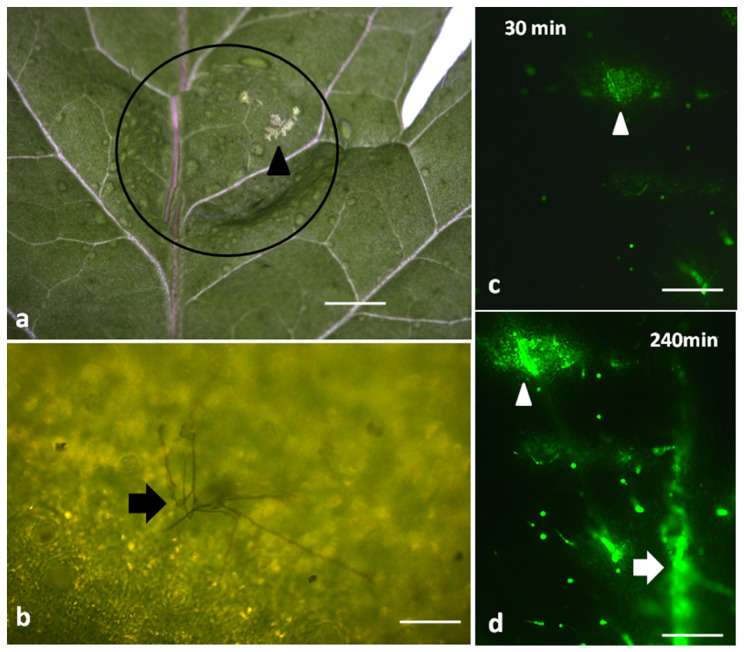
A visible damage (arrowhead) can be observed on a tomato leaf 24 h after application of a *Botrytis cinerea* spore suspension drop (circle) (**a**). *B. cinerea* hyphae (arrow) visible on the leaf epidermis 24 h after the spore suspension application (**b**). Staining of tomato leaflets with 2′, 7′-dichlorofluorescein diacetate (DCF-DA) for hydrogen peroxide (H_2_O_2_) detection after 30 (**c**) and 240 min (**d**) of *B. cinerea* application. Hydrogen peroxide generation is visible as green fluorescence (**c**,**d**). Scale Bars: 1 mm in (**a**), 0.2 mm in (**b**), and 500 μm in (**c**,**d**).

**Figure 2 plants-10-00521-f002:**
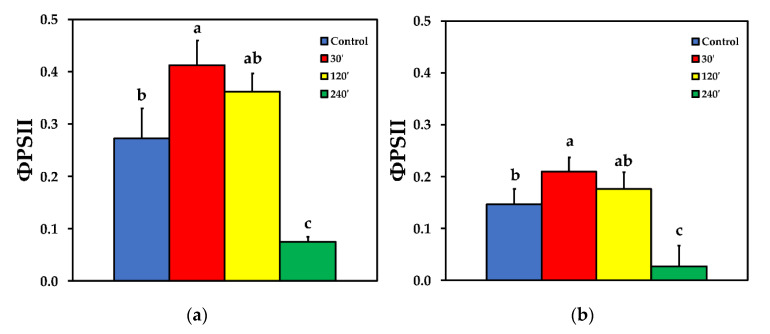
The effective quantum yield of PSII photochemistry (Φ_PSII_) before (control) and after 30, 120 and 240 min of *B. cinerea* aquatic spore suspension application on tomato leaflets, measured at 230 μmol photons m^−2^ s^−1^ (**a**), and at 900 μmol photons m^−2^ s^−1^ (**b**). Error bars are standard deviations (n = 6). Columns with different letters are statistically different (*p* < 0.05).

**Figure 3 plants-10-00521-f003:**
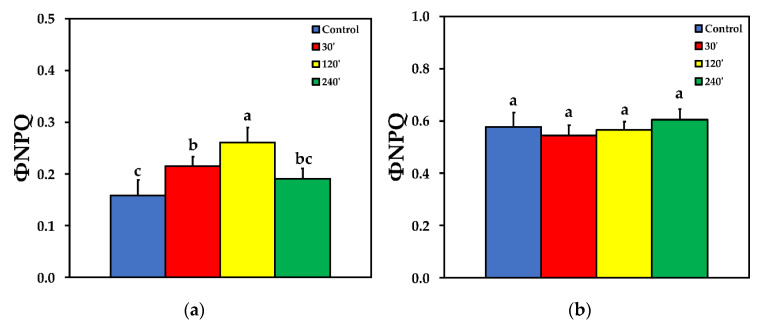
The quantum yield of regulated non-photochemical energy loss in PSII (Φ_NPQ_) before (control) and after 30, 120 and 240 min of *B. cinerea* aquatic spore suspension application on tomato leaflets, measured at 230 μmol photons m^−2^ s^−1^ (**a**), and at 900 μmol photons m^−2^ s^−1^ (**b**). Error bars are standard deviations (n = 6). Columns with different letters are statistically different (*p* < 0.05).

**Figure 4 plants-10-00521-f004:**
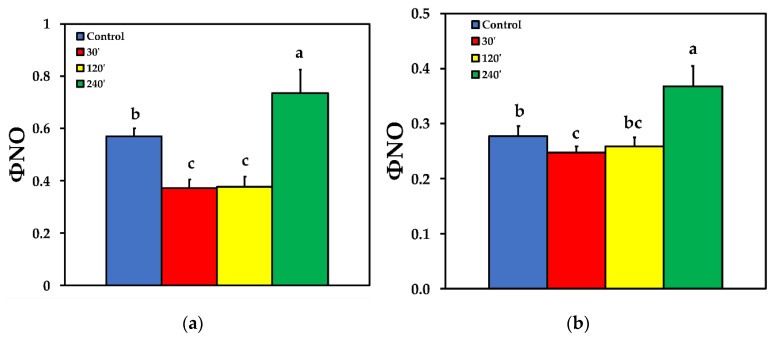
The quantum yield of non-regulated energy loss in PSII (Φ_NO_) before (control) and after 30, 120 and 240 min of *B. cinerea* aquatic spore suspension application on tomato leaflets, measured at 230 μmol photons m^−2^ s^−1^ (**a**), and at 900 μmol photons m^−2^ s^−1^ (**b**). Error bars are standard deviations (n = 6). Columns with different letters are statistically different (*p* < 0.05).

**Figure 5 plants-10-00521-f005:**
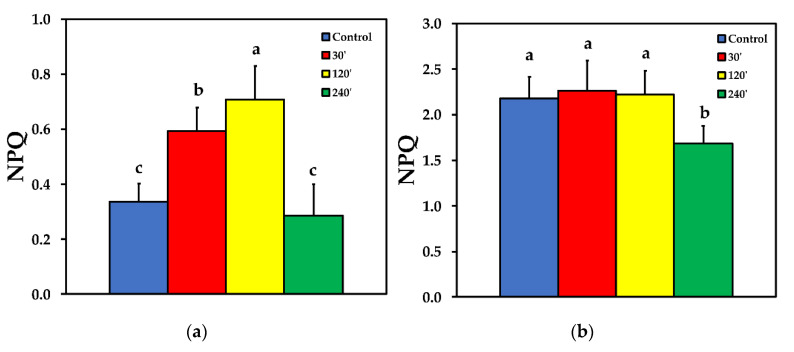
The non-photochemical quenching (NPQ), reflecting heat dissipation, before (control) and after 30, 120 and 240 min of *B. cinerea* aquatic spore suspensions application on tomato leaflets, measured at 230 μmol photons m^−2^ s^−1^ (**a**), and at 900 μmol photons m^−2^ s^−1^ (**b**). Error bars are standard deviations (n = 6). Columns with different letters are statistically different (*p* < 0.05).

**Figure 6 plants-10-00521-f006:**
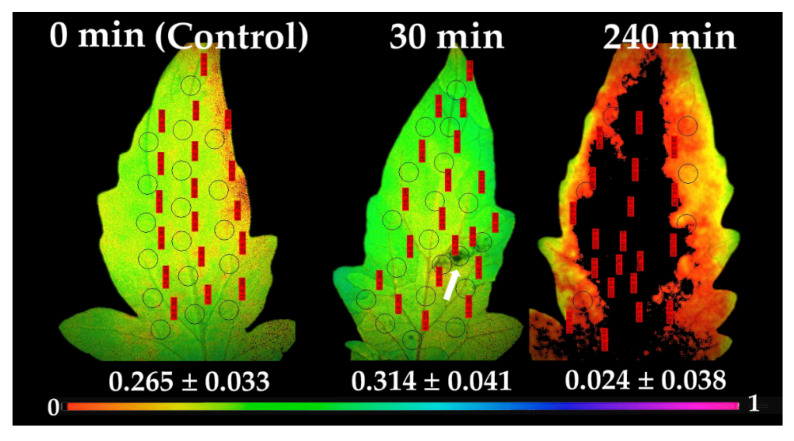
Chlorophyll fluorescence images of Φ_PSΙΙ_, measured at 640 μmol photons m^−2^ s^−1^, before (0 min, control) and after 30 and 240 min of *B. cinerea* suspension application on tomato leaflets. The color code depicted at the bottom ranges from values 0.0 to 1.0. The circles in each image denote the areas of interest (AOI) that were selected in each leaflet and are complemented by red labels with their Φ_PSΙΙ_ values, while whole leaflet value is presented. Arrowhead at Φ_PSΙΙ_ image after 30 min of *B. cinerea* suspension application points at the visible area of the aquatic spore suspensions.

**Figure 7 plants-10-00521-f007:**
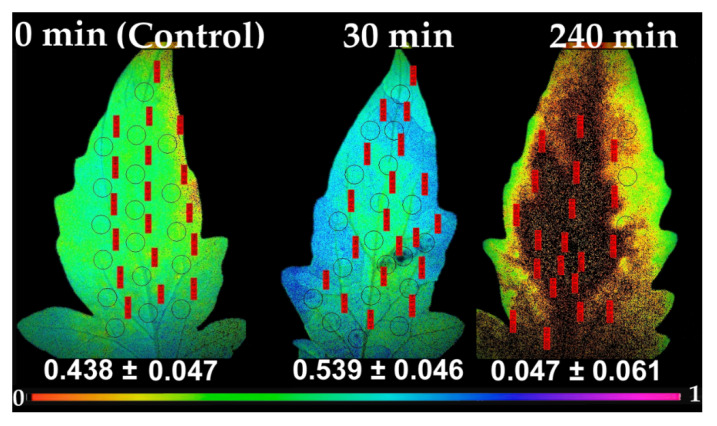
Chlorophyll fluorescence images of the fraction of open PSII reaction centers (*q*_p_), measured at 640 μmol photons m^−2^ s^−1^, before (0 min, control) and after 30 and 240 min of *B. cinerea* spore suspension application on tomato leaflets. The color code depicted at the bottom ranges from values 0.0 to 1.0. The circles in each image denote the areas of interest (AOI) that were selected in each leaflet and are complemented by red labels with *q*_p_ values, while whole leaflet value is presented.

**Figure 8 plants-10-00521-f008:**
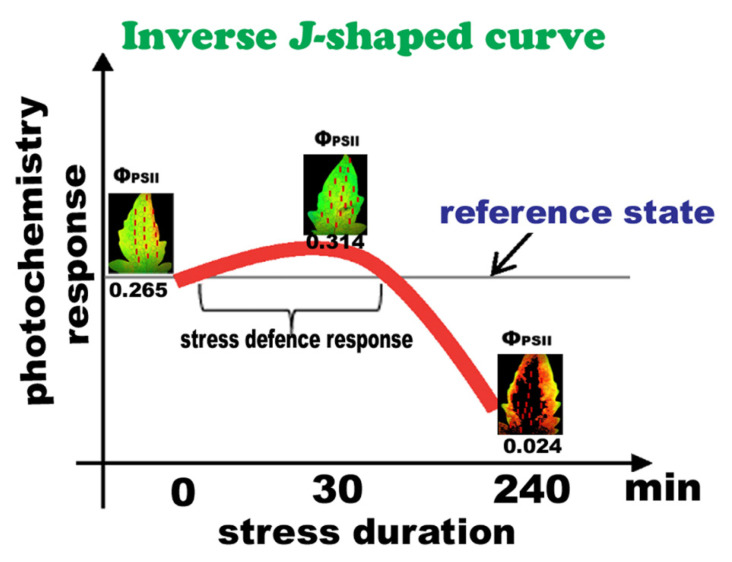
Overview of the hormetic response of photosystem II in tomato (at moderate light intensity), to *B. cinerea* spore application. The hormetic effect is defined by an inverse *J*-shaped biphasic curve with a short exposure time to have a stimulatory effect but at longer exposure time severe toxic effects to be apparent.

**Table 1 plants-10-00521-t001:** Definitions of the chlorophyll fluorescence parameters and their calculation.

Parameter	Definition	Calculation
Φ_PSII_	The effective quantum yield of PSII photochemistry	(Fm′ − Fs)/Fm′
Φ_NPQ_	The quantum yield of regulated non-photochemical energy loss in PSII, that is heat dissipation for photoprotection	Fs/Fm′ − Fs/Fm
Φ_NO_	The quantum yield of non-regulated energy loss in PSII	Fs/Fm
NPQ	The non-photochemical quenching that reflects heat dissipation of excitation energy	(Fm − Fm′)/Fm′
*q* _p_	The photochemical quenching, that is the redox state of the plastoquinone pool, is a measure of the number of open PSII reaction centers	(Fm′ − Fs)/(Fm′ − Fo′)

## Data Availability

No new data were created or analyzed in this study. Data sharing is not applicable to this article.

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
