# Peer review of "Hormetic Responses of Photosystem II in Tomato to Botrytis cinerea"

_plants, 2021, doi:10.3390/plants10030521_

Round 1

Reviewer 1 Report

Dear authors, your work contains interesting results, but I do not agree with their interpretation. In my opinion, the justification of the theory of hormesis in your manuscript is insufficient. Please, change the interpretation of the results, focus on the important observation you made, mainly the rapid reaction of the photosystem to the Bortytis infection

If you still believe that the photosystem response meets the criteria of hormesis, then you must complete the results and provide the value of the stress factor dose.

Author Response

Dear authors, your work contains interesting results, but I do not agree with their interpretation. In my opinion, the justification of the theory of hormesis in your manuscript is insufficient. Please, change the interpretation of the results, focus on the important observation you made, mainly the rapid reaction of the photosystem to the Bortytis infection

If you still believe that the photosystem response meets the criteria of hormesis, then you must complete the results and provide the value of the stress factor dose.

We have answered to your comment by including in the new section Conclusions, documentation regarding the features of hormesis as a function of time. The temporal features of hormesis are recognized in the article (Dose-Response 2019, 17, 1–3) [36], in a recent review (Trends Plant Sci. 2020, 25, 1076-1086) [65] and in the recently published article of our group (Int. J. Mol. Sci. 2021, 22, 41) [13]. Reference to more published studies with higher plants and algae reporting a time-dependent hormesis can be found in the abovementioned review.

Title: In my opinion the title is not adequate for the information included in the publication. The title of the work should be formulated precisely and unambiguously. Hormotic responses are not well described in the introduction and your results don’t document this theory well. In your paper, Botrytis cinerea is the only one factor causing biotic stress and this should be emphasized in the title.

The title has been changed accordingly.

The paper presents the results of the research of the effect of Botrytis spore infection  on chlorophyll fluorescence in tomato leaves. The swift response of the photosystem II to the grey mold infection  has been documented and, in my opinion, this is a very important observation.

In my opinion, the Introduction, Results, Materials, Methods and Bibliography are well-written. You present many results typical for photochemistry, but interpret them in the light of the theory of hormesis.

If I understood correctly your interpretation, according to hormesis, endogenous hydrogen peroxide is formed in a leaf under the influence of fungal infection. The process occurs first in a low concentration H2O2, which stimulates a defensive reaction. Then concentration of H2O2 (and other ROS) increases so that the free radicals destroy the photosystem and cause necrosis of the leaves. Your interpretation of the results may be true but is not sufficiently documented and is one-sided.

line 176-186.

We have re-written the paragraph.

During infection, hydrogen peroxide can be produced by both, Botrytis and the leaf of tomato. The data indicate that the effect of fungal infection on plant  and the  response of the plant to pathogen attack depend on both partners as well as on the experimental conditions.  B. cinerea a presumably secrete enzymes to breach the plants’ surface. There are enzymes that generate reactive oxygen species (ROS), for example superoxide dismutase BcSOD1 produces hydrogen peroxide  via superoxide anion dismutation reactions. B. cinerea produces ROS in axenic culture and in planta, cytochemical analysis showed the presence of in hyphal tips and H2O2 generation in and around the penetrated cell wall [DOI: 10.1046/J.1364-3703.2004.00201.X]. Another source of hydrogen peroxide in Botrytis infection is  the toxin botrydial. All virulent strains of B. cinerea have produced botrydial, both in vitro and in planta. Colmenares et al., 2002 J. Chem. Ecol.28, 997–1005 shows that botrydial displays its toxic effect (when applied externally on the leaves of bean plants) only in the presence of light. Also, according to the author, botrydial decomposes to botryendial and hydrogen peroxide. The role of hydrogen peroxide as an indicator in plants is well known. In plant cells, there are antioxidant enzymes and antioxidants that prevent the harmful effects of ROS. RFT  affect signaling pathways and the expression of many genes. They are also involved in the processes of apoptosis. As you observe the increase in fluorescence, it may indicate both production of H2O2 during the  stress defense mechanism of plant cells  and ROS secreted by Bortytis.

You have not studied what genes of the plant cells are expressed after infection with Botrytis. Therefore, it cannot be concluded that the retrograde regulation route is being launched.

Line -195- 198

We have re-written the text incorporating in the discussion the suggested citations [31,33] and withdrawing the phrase “retrograde regulation”.

You didn’t measure the quantity of hydrogen peroxide in the leaf.  You just found that its concentration was increasing. Photosystem dysfunction is related to the generation of ROS, but we do not know what are the ROS concentration  in the leaf  or whether  are ROS sufficient for cell death. The damage to the chloroplast does not always mean cell death.

Line 222-225

We don’t have any quantity measurements of leaf hydrogen peroxide but we draw our conclusion based on the quantum yield measurements (lines 156-159, ΦPSII=0) (Figure 6) and on the fraction of open reaction centers (lines 163-166) presented in Figure 7 (qp=0). We changed the sentence from “However, 240 min after B. cinerea application the high levels of 1O2 produced (Figure 4) activated a programmed cell death [55] that acted synergistically with the high H2O2 level (Figure 1d), resulting in the tomato leaf necrosis (Figures 6,7) [15].” to (lines 226-230) “However, 240 min after B. cinerea application, the high levels of 1O2 produced (Figure 4) could acted synergistically with the high H2O2 level (Figure 1d), resulting in chloroplast damage [15,55]. Botrytis cinerea is a necrotrophic fungus that produces constantly toxic compounds which ultimately cause cell death [4]. Then, the fungus feeds on the dead tissue, thus, causing noticeable necrotic lesions [4].”

The paper does not present the results confirming the occurrence of apoptosis at the infection site, ΦPSII damage does not mean cell death. In leaves of Arabidopsis, Quilliam [doi:10.1093/jxb/erj039 ] reported an increase in ΦPSII after 24h of being infected. The cells with reduced value of ΦPSII died or fully regenerated.

Line 234-235,  253-254

We do not report reduced ΦPSII values but as we mentioned before we report zero quantum yield and completely closed reaction centers. However, the conclusions reached were changed to: “chloroplast damage”, “leaf damage” and “damaging to PSII functionality”.

Please, explain the relationship between jasmonic acid or β-aminobutyric acid (BABA) and the parameters determined in the study.

Line 29-30  Line 240-254

The relationship refers to the defence response as it is re-written in lines 251-253.

For me, the interpretation of the results is unclear. Your explanation the mechanism of hormesis is not clear. Hormesis is a dose–response phenomenon that is characterized by low-dose stimulation and high-dose inhibition. I have not found  a quantitative evaluation of the hormetic dose response. In my opinion, you cannot put the duration of stress on the OX axis, but the dose of the factor.

The temporal features of hormesis are recognized in a recent review (Trends Plant Sci. 2020, 25, 1076-1086) [65] and in the recently published article of our group (Int. J. Mol. Sci. 2021, 22, 41) [13]. Reference to more published studies with higher plants and algae reporting a time-dependent hormesis can be found in the abovementioned review.

In the new section Conclusions, we present documentation regarding the features of hormesis as a function of time. It is written: “There are published studies reporting a time-dependent hormesis [36,65], but it is only recently that analysis has been made regarding the features of hormesis as a function of time [13,64,65]”.

Reviewer 2 Report

In the manuscript "Hormetic Responses of Photosystem II Photochemistry in Tomato to Biotic Stress”, Stamelou and collaborators studied the impact of B. cinerea pathogen on photosystem II (PSII) photochemistry of tomato leaves by using chlorophyll fluorescence imaging analysis, trying to understand the mechanisms involved in plant defense response to the fungal pathogen. The authors showed increased levels of hydrogen peroxide and enhanced PSII functionality after 30 min of B. cinerea application, being then PSII inhibited after longer exposure to B. cinerea spores. The authors concluded that stronger plant mechanisms may be activated through the induction of defense genes and a retrograde signaling mechanism may be present giving tolerance against B. cinerea at short-time exposure.

The manuscript is very well written and well organized. The abstract is concise, clear and presents an overview of the results obtained. The Introduction provides a good, generalized background of the topic. The experimental design is correct, the methodologies are well described and appropriate for the objectives of the work.  The Results section presents clearly the results and these are well interpreted. The graphics and images are well and clearly presented. The authors make a good discussion of the results and these are compared with previous data.

However, I feel that, even though, the conclusions are presented in the Abstract, a final paragraph in the Discussion section is missing with the main conclusions of the study.

Author Response

In the manuscript "Hormetic Responses of Photosystem II Photochemistry in Tomato to Biotic Stress”, Stamelou and collaborators studied the impact of B. cinerea pathogen on photosystem II (PSII) photochemistry of tomato leaves by using chlorophyll fluorescence imaging analysis, trying to understand the mechanisms involved in plant defense response to the fungal pathogen. The authors showed increased levels of hydrogen peroxide and enhanced PSII functionality after 30 min of B. cinerea application, being then PSII inhibited after longer exposure to B. cinerea spores. The authors concluded that stronger plant mechanisms may be activated through the induction of defense genes and a retrograde signaling mechanism may be present giving tolerance against B. cinerea at short-time exposure.

The manuscript is very well written and well organized. The abstract is concise, clear and presents an overview of the results obtained. The Introduction provides a good, generalized background of the topic. The experimental design is correct, the methodologies are well described and appropriate for the objectives of the work. The Results section presents clearly the results and these are well interpreted. The graphics and images are well and clearly presented. The authors make a good discussion of the results and these are compared with previous data.

However, I feel that, even though, the conclusions are presented in the Abstract, a final paragraph in the Discussion section is missing with the main conclusions of the study.

Thank you for your positive comments. Taking into account your comment we added a new section, Conclusions, with a paragraph in it. 

Reviewer 3 Report

This MS is about the effect of Botrytis cinerea infection on photosynthetic activity of tomato plants. 

The MS is almost well designed and written, but I have some major and minor comments:
- How were the plants treated? Have the authors used detached leaves or in vivo plants for the treatments? How many leaves were treated on one plant? How old were the treated leaves? I can see droplets on the surface of the leaf in the fig 1a, is that water? If it is water, was it applied before the infection or later? If later, how can the authors prove that it does not dilute or wash the inoculum?
- Moreover, from the scale bars, one can think that there is only about 1.8 magnification but seems impossible for me.
- In my opinion, the lesion that can be seen in fig 1a cannot be formed in four hours with the amount and size of hyphae shown in fig 1b. I think it takes at least 24-48 hours after the infection to form that size of the lesion.
- What was the reason for using different light intensities? Again, I think, the high light intensity (compared to the growing conditions) alone can cause this kind of oxidative stress response. Why the results of ML are not showed in the graphs? Why the images of the LL and HL are not showed, what was the reason for choosing the ML to show the images created with the Imaging PAM?
- How can the authors certify the decreased amount of singlet oxygen formation?

Author Response

Τhis MS is about the effect of Botrytis cinerea infection on photosynthetic activity of tomato plants.

The MS is almost well designed and written, but I have some major and minor comments:

We appreciate comments raised by the reviewer to each of which we provide an answer below.

- How were the plants treated? Have the authors used detached leaves or in vivo plants for the treatments? How many leaves were treated on one plant? How old were the treated leaves? I can see droplets on the surface of the leaf in the fig 1a, is that water? If it is water, was it applied before the infection or later? If later, how can the authors prove that it does not dilute or wash the inoculum?

A detail description answering these questions has been included in materials and methods (lines 267-272 and 298-300) as follows: “Tomato 5th leaflets were detached and laid on Petri dishes containing two sterilized blotting filter papers (Whatman, Schleicher and Schuell, Ottawa, Canada), wetted with sterile water. A drop (20 μl) of B. cinerea aqueous spore suspension (105-107 spores/ml) was applied on tomato leaflets with a micropipette [62] as shown on Figure 1a, while leaves treated with a drop (20 μl) of sterile water (without B. cinera spores) were considered as controls”, and “Statistically significant differences among the means were determined, from three independent treatments with each treatment consisting of three leaflets from three different tomato plants, using two-way ANOVA.”

- Moreover, from the scale bars, one can think that there is only about 1.8 magnification but seems impossible for me.

Scaling has been corrected accordingly. Thank you for pointing this.

- In my opinion, the lesion that can be seen in fig 1a cannot be formed in four hours with the amount and size of hyphae shown in fig 1b. I think it takes at least 24-48 hours after the infection to form that size of the lesion.

Yes, you are right, figures 1a and 1b represent observation after 24 hours of spore suspension application, but by mistake the time was written 240 minutes. Now it has been corrected. Thank you for pointing this.

- What was the reason for using different light intensities? Again, I think, the high light intensity (compared to the growing conditions) alone can cause this kind of oxidative stress response. Why the results of ML are not showed in the graphs? Why the images of the LL and HL are not showed, what was the reason for choosing the ML to show the images created with the Imaging PAM?

We performed chlorophyll fluorescence measurements with the actinic light intensities of 230 μmol photons m−2 s−1 (LL) similar to the growth light, of 640 μmol photons m–2 s–1 (ML), and of 900 μmol photons m−2 s−1 (HL). However, images were captured only at the ML intensity of 640 μmol photons m–2 s–1. The results of ML are not shown in the graphs in order to avoid double presentation of the same results. By using LL and HL intensities we wanted to check if HL increases the effects or not. We observed that (lines 196-198) “when Botrytis application was combined with HL, the photoprotective mechanism was no longer buffering the excess light stress levels, indicating an imbalance between energy supply and demand”.

The measurements of tomato leaflets with B. cinerea aqueous spore suspension at HL treatment are compared with control measurements of tomato leaflets (without B. cinera spores) at HL treatment also, in order to observe any differences. 

- How can the authors certify the decreased amount of singlet oxygen formation?

We estimate the decreased amount of singlet oxygen formation from (lines 220-225) “the significant decrease in the quantum yield of non-regulated energy loss in PSII (ΦNO) (Figure 4). ΦNO comprises of chlorophyll fluorescence internal conversions and intersystem crossing, that results to 1O2 formation via the triplet state of chlorophyll (3chl*) [34,48-50]. Thus, after application of B. cinerea on tomato leaflets, there was a decrease of 1O2 that is considered as a highly damaging ROS produced by PSII [14,51-54]”.

Round 2

Reviewer 1 Report

Dear authors, thank you for the explanation and supplementation of the manuscript. I appreciate that you provided additional literature citations and sustained your interpretation of the results.

In my opinion, the publication is now factually correct and valuable. Good luck in your further research.

Author Response

Dear authors, thank you for the explanation and supplementation of the manuscript. I appreciate that you provided additional literature citations and sustained your interpretation of the results.

In my opinion, the publication is now factually correct and valuable. Good luck in your further research.

Thank you for your positive comments.

Reviewer 3 Report

I agree with most of the comments, but I still do not understand or I missed its explanation of why the authors use the 640 μmol photons m − 2 s – 1 light intensity. I did not use the Imaging PAM for ten years now, but I can still remember that we made photos that were analyzed later only, so I still have all the images taken from those experiments. So it would be nice to see those images that were later analyzed (LL or HL) in these cases also. I can imagine that there is no difference between the LL and ML results, but this part is difficult to explain.

The decrease of singlet oxygen formation is only an estimation, please try to redefine the sentence with the phrase: there is a possibility of singlet oxygen decrease regarding the measured changes in  ΦNO parameter.

My other comment, which I missed mentioning last time is about the graphs. Please use patterns in case of the bars instead of filling them out with different colours.

Author Response

I agree with most of the comments, but I still do not understand or I missed its explanation of why the authors use the 640 μmol photons m − 2 s – 1 light intensity. I did not use the Imaging PAM for ten years now, but I can still remember that we made photos that were analyzed later only, so I still have all the images taken from those experiments. So it would be nice to see those images that were later analyzed (LL or HL) in these cases also. I can imagine that there is no difference between the LL and ML results, but this part is difficult to explain.

As we mentioned before we performed chlorophyll fluorescence measurements with the actinic light intensities of 230 μmol photons m−2 s−1 (LL), 640 μmol photons m–2 s–1 (ML), and 900 μmol photons m−2 s−1 (HL). When we save chlorophyll fluorescence data, we save first all the measured chlorophyll fluorescence parameters simultaneously in excel files in the light intensity that we measure, and then for the images we have to save them for every chlorophyll fluorescence parameter separately in the light intensity that we measure. Since we wanted to measure the same leaf 30 min, 120 min and 240 min after spore suspension application we did not have enough time between the measurements to save all the images at the three light intensities, thus we saved the images at 640 μmol photons m–2 s–1 only, while the measurements were saved in excel files for all the measured chlorophyll fluorescence parameters in the three light intensities. Thus, images were captured only at the ML intensity of 640 μmol photons m–2 s–1. Consequently, we did not analyse photos later to extract the data but the measurements were saved separately in excel files before saving the images.

The results of ML are not shown in the graphs in order to avoid double presentation of the same results. By using LL and HL intensities we wanted to check if HL increases the effects or not. We observed that (lines 196-198) “when Botrytis application was combined with HL, the photoprotective mechanism was no longer buffering the excess light stress levels, indicating an imbalance between energy supply and demand”.

The decrease of singlet oxygen formation is only an estimation, please try to redefine the sentence with the phrase: there is a possibility of singlet oxygen decrease regarding the measured changes in ΦNO parameter.

We have re-written the text taking into account your suggestion (lines 25-26 and 225-227).

My other comment, which I missed mentioning last time is about the graphs. Please use patterns in case of the bars instead of filling them out with different colours.

We are using different colours in the bars of the graphs for manuscripts submitted in MDPI journals that do not have extra fees for colour figures, but for other journals that demand extra fees for colour figures we use black and white patterns.